materials science/nanotechnology/synthetic chemistry

organic modified montmorillonite, amphiphilic polymer nanocomposite, hydrophobic association, rheology

**Author for correspondence:**
Yangchuan Ke
e-mail: kyc031@sohu.com

This article has been edited by the Royal Society of Chemistry, including the commissioning, peer review process and editorial aspects up to the point of acceptance.

# Preparation and properties of amphiphilic hydrophobically associative polymer/montmorillonite nanocomposites

Cancan Bai, Yangchuan Ke, Xu Hu, Liang Xing, Yi Zhao, Shichao Lu and Yuan Lin

CNPC Nanochemistry Key Laboratory, College of Science, China University of Petroleum, Beijing 102249, People's Republic of China

YK, 0000-0002-2989-1551

In this research, a novel amphiphilic hydrophobically associative polymer nanocomposite (ADOS/OMMT) was prepared using acrylamide (AM), sodium 4-vinylbenzenesulfonate (SSS), N, N′-dimethyl octadeyl allyl ammonium bromide (DOAAB) and organo-modified montmorillonite (OMMT) through *in situ* polymerization. Both X-ray diffraction patterns and transmission electron microscopy images verified the dispersion morphology of OMMT in the copolymer matrix. Then, the effect of the introduction of OMMT layers on the copolymer properties was studied by comparing with pure copolymer AM/SSS/DOAAB (ADOS). The thermal degradation results demonstrated that the thermal stability of the ADOS/OMMT were better than pure copolymer ADOS. During the solution properties tests, ADOS/OMMT nanocomposite was superior to ADOS in viscosifying ability, temperature resistance, salt tolerance, shear resistance and viscoelasticity, which was because OMMT contributed to enhance the hydrophobic association structure formed between polymer molecules. Additionally, the ADOS/OMMT nanocomposite exhibited more excellent interfacial activity and crude oil emulsifiability in comparison to pure copolymer ADOS. These performances indicated ADOS/OMMT nanocomposite had good application prospects in tertiary recovery.

## 1. Introduction

On one hand, oil resources are being consumed rapidly owing to the population growth and the enhancement of living quality, and

on the other hand, with the increasing of difficulties for exploiting oil and gas, the primary and secondary recovery techniques are being phased out, which is driving the development of enhanced oil recovery technologies [1,2]. Over the last few decades, a variety of extraction methods have emerged, among which chemical flooding is one of the most widely used methods to enhanced oil recovery (EOR) [3]. Polymer flooding, polymer-surfactant (Sp) flooding and polymer-surfactant-alkali are several commonly applied and effective methods of chemical EOR [4]. Of them, polyacrylamide (PAM) and partially hydrolysed polyacrylamide (HPAM) are extensively used as water-soluble polymers to correct the water/oil mobility ratio owing to their excellent viscosification properties and relatively lower cost [5,6]. Nevertheless, the viscosity of PAM and HPAM solutions tends to decrease sharply in extreme environments with high-salinity, high-temperature and high-shear rates [7,8]. In order to cope with the challenges brought by these harsh conditions in EOR application, hydrophobically associating polymers, because of their unique properties, have attracted the attention of many researchers at home and abroad in recent decades [9–11].

Hydrophobic associating polymer is a kind of water-soluble polymer with a few hydrophobic groups in the hydrophilic main chain [12,13]. When the concentration of polymer solution is not less than the critical association concentration (CAC) value, the hydrophobic groups which are blocky or randomly distributed on the molecule backbone tend to form a reversible supermolecular network in aqueous solution by intermolecular hydrophobic association [14–16]. Therefore, compared with traditional polymers, hydrophobic associating polymer solution behaves with better rheological properties, such as strong salinity tolerance, good shearing resistance and temperature resistance [1,17]. In addition, hydrophobic associating polymers usually present interfacial activity and emulsification properties owing to the introduction of hydrophobic parts [15,18]. However, often in literature about hydrophobic association polymers, researchers usually pay more attention to the effect of hydrophobic association or functional structure on the solution properties of polymers, while the studies of interface properties and emulsification properties are seldom considered [19].

In recent years, the research of polymer nanocomposites has become the hot topic of the materials science and petrochemicals area because of their extraordinary application potential in diverse fields [20,21]. Polymer nanocomposites can be obtained by incorporating nanoscale inorganic materials into host polymer [22,23]. Among the various inorganic nanometer materials, such as layered silicates, nanotubes, nanofibres, spherical nanoparticles like silica and nano-ZnO; sodium-montmorillonite (Na-MMT) as a one of the natural layered silicates is very widely selected to make polymer-based nanocomposite owing to its intercalation/exfoliation characteristics, large specific surface, high aspect ratio, great ionic exchange capacity, low cost and good environmental benefit [24–26]. Polymer/MMT nanocomposites can achieve more excellent thermal stability, mechanical, wettability and barrier properties relative to pure polymers on account of strong interfacial interactions generated between the dispersed MMT platelets and the polymer matrices [27–30]. The dispersion quality of MMT platelets in the polymer matrix is an important factor impacting the formation of effective interfacial interaction between them [31–34]. Good dispersion and exfoliation of MMT in polymer matrix is the prerequisite for significant improvement of nanocomposite properties [35–37]. However, natural Na-MMT is hydrophilic because of the presence of $Na^+$ and $K^+$ ions between its layers. Therefore, in order to increase the compatibility of MMT platelets with hydrophobic polymer bulk, the original MMT needs to carry out organic pre-modification by cation exchange technology [34,38]. Many previous studies have indicated the organic MMT is easier for achieving homogeneous dispersion and exfoliation in the polymer matrix [36,39]. In addition, *in situ* polymerization is a frequently used polymerization process for preparing nanocomposites. Because the chain propagation reaction of the polymer can occur directly between the MMT interlayers, the reaction heat can be generated during the process, both of which promote the exfoliation and dispersion of the MMT in the polymer matrix [35,40]. Up to now, much research has been performed on polymer/montmorillonite nanocomposites, especially about thermal, optical and mechanical properties. Recently, several researchers in the field of petroleum chemistry have studied the effects of MMT nanoparticles on the properties of polymer solutions used for EOR technologies. For example, Rezaei *et al.* found that the temperature resistance, salts tolerance and shear resistance of HPAM aqueous solutions were enhanced by adding modified clay nanoparticles (SMCN). Compared with the ordinary polymer flooding processes, SMCN-HPAM improved oil recovery of about 33% [41]. Hu *et al.* synthesized β-cyclodextrin-functionalized polyacrylamide/MMT nanocomposites, which exhibited better temperature resistance, salts tolerance, rheological properties, thermal stability and higher viscosity than pure polymer [42]. However, research on properties of aqueous solution of water-soluble amphiphilic hydrophobically associative polymer/modified MMT nanocomposites have rarely been reported previously.

**Figure 1.** Scheme for the preparation of (*a*) DOAAB and (*b*) OMMT.

In this article, we successfully synthesized a amphiphilic hydrophobically associative polyacrylamide nanocomposite through *in situ* polymerization. Fourier transform infrared (FTIR), [1]hydrogen nuclear magnetic resonance ([1]HNMR), transmission electron microscopy (TEM) and X-ray diffraction (XRD) were employed for the characterization of structure and morphology of the copolymer/organo-modified montmorillonite (OMMT) nanocomposites and pure polymer. In addition, a thermo-gravimetric analyser (TGA) was applied for the measurement of thermal property of these samples. The solution properties, interfacial activities and emulsification ability of the copolymer/OMMT nanocomposites were studied, which are compared to those of pure polymers.

# 2. Material and experiments

## 2.1. Materials

Acrylamide (AM), sodium p-styrene sulfonate hydrate (SSS) and 2, 2′-Azobis [2-(2-imidazolin-2-yl) propane] dihydrochloride (AIBI) were produced by Shanghai Macklin Biochemical Co., Ltd. Octadecyl dimethyl tertiary amine were obtained from Shandong Xiya Chemical Industry Co., Ltd. Allyl bromide was bought from Aladdin Industrial Corporation (China). CTAB was obtained from Tianjin Guangfu Fine Chemical research institute. Na-MMT with a cationic exchange capacity of $1 \, mol \, l \, kg^{-1}$ was bought from Huaian Saibei Technology Co. Ltd. Sodium chloride (NaCl), anhydrous diethyl ether, acetone and ethanol were purchased from Beijing chemical works. Anhydrous calcium chloride ($CaCl_2$) was provided by Fuchen (Tianjin) Chemical Reagents Co., Ltd. The crude oil was come from Panyu, Guangdong province (its density was $0.8101 \, g \, cm^{-3}$). Deionization water was purchased in Beijing at Yuerui Huaqiang Science and Technology Ltd. All reagents above were used directly without further purification, because they were analytical grade.

## 2.2. Synthesis of dimethyl octadeyl allyl ammonium bromide

First, 1000 mmol N, N-dimethyl-octadecyl tertiary amine was dissolved in a three-mouth flask containing 300 ml of ethanol, and 150 mmol allyl bromide was added to the three-mouth flask for stirring. Then, the reaction was conducted under the protection of nitrogen at 60°C for 24 h. Third, the reaction mixture was distilled under reduced pressure to remove the ethanol thereby obtaining the crude product. Finally, the crude product needed to be recrystallized using acetone and ether and then put in the refrigerator for 48 h. The solvent in the product was filtered out to obtain the pure white powder product, which was put into the oven to dry for later use. The synthetic scheme is demonstrated in figure 1*a*.

## 2.3. Preparation of organic intercalated modified montmorillonite

OMMT was obtained by cation exchange reaction, and the details are as follows. Firstly, 5 g pure natural Na-MMT powders was added into 100 ml of deionized water in a flat bottom three-mouth flask. Then, the mixture was stirred continuously for 1.5 h at 80°C in a water bath. At the same time, 1.2 CEC

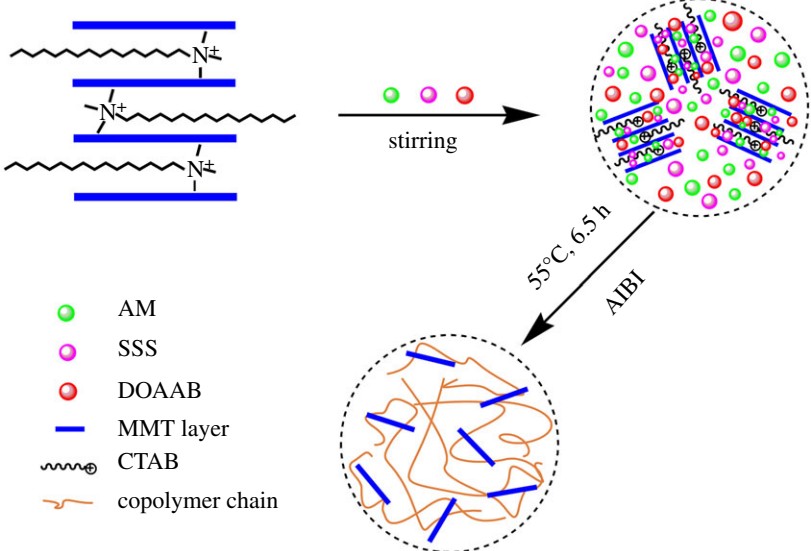

**Figure 2.** *In situ* polymerization for the preparation of ADOS/OMMT nanocomposites.

intercalation agent (CTAB) was dissolved in 40 ml of deionized water in a 150 ml beaker for later use. Finally, intercalation agent solution was slowly dripped into the swollen montmorillonite dispersion system above, and the dispersion system continued to react for 10 h at 80°C with stirring. After the intercalation reaction, the prepared suspension of OMMT was allowed to stand for 12 h in a separatory funnel. Then, the upper parts of samples were collected by filtering and lavation was carried out several times with deionized water until no Br⁻ ion could be detected by solution of AgNO₃. The finished samples were dried for one day in a drying oven at 70°C, dry samples were ground to powder (around 65∼75 µm) and put into self-sealing bags for later use. The process of organic modification of MMT is demonstrated in figure 1*b*.

## 2.4. Synthesis of ADOS/organic intercalated modified montmorillonite nanocomposites

*In situ* polymerization was used for synthesizing ADOS/OMMT nanocomposites. In a 150 ml beaker, 12 g AM, 0.42 g N′-dimethyl octadeyl allyl ammonium bromide (DOAAB), and 0.385 g SSS were dissolved in 40 ml deionization water in sequential order forming a uniform solution. After that the different quantities of prepared OMMT (1% and 2% relative to total mass of monomers, respectively) were added into the above monomers solution slowly under high speed magnetic stirring and kept stirring for 2.5 h at 40°C. This was done to make sure that monomers adsorbed onto the MMT nanolayers adequately to synthesize composite material with better performance. An initiator AIBI (0.25 wt%) was added into the mixed suspension by injector, then, the mixture was stirred for another 30 min. The two steps were done under nitrogen protection. Subsequently, copolymerization was carried out for 6.5 h at 55°C. The resulting gel copolymer was cut into small pieces and washed in ethyl alcohol repeatedly, and dried them in a vacuum oven at 75°C. The scheme for the preparation of ADOS/OMMT nanocomposite is demonstrated in figure 2.

Pure copolymer (ADOS) was synthesized by almost the same process and condition. The only difference was that OMMT was not needed in the synthesis process.

## 2.5. Characterization

FTIR spectrometric analysis in the spectral range of 4000–400 cm⁻¹ was carried out for OMMT, Na-MMT, pure copolymer and nanocomposite by a FTS-3000 spectrophotometer (American Digilab) at room temperature. The ¹H-NMR spectra of pure copolymer and surfactant monomer was measured by a Bruker ASCEND-400 NMR instrument, and deuterium oxide (D₂O) was used as solvent. The interlayer spacing of the samples was characterized on a Bruker D8 Advance X-ray diffraction instrument (STADIP, Germany) and calculated by Bragg's Law. During the testing, the system

operated at a voltage of 40 KV and current 40 mA. Also the scanning speed was $1° \min^{-1}$ from 1° to 10° with a step distance of 0.02°.

The weight losses of the samples from room temperature to 700°C were recorded on a NETZSCH TGA (STA409PC) from Germany, and the samples were heated at the rate of $10°C \min^{-1}$ under nitrogen flow rate of $140 \, cm^3 \min^{-1}$. Dispersion morphology of the OMMT layers in the nanocomposite was observed by TEM (F20, Japan) at 200 kV. Meanwhile, the microscopic observation of pure copolymer matrix was obtained as blank control.

## 2.6. Determination of intrinsic viscosity

The intrinsic viscosity of copolymers and nanocomposites was obtained by the dilution extrapolation method at 30°C. First, the time of the sample solution with NaCl concentration of $1 \, mol \, l^{-1}$ flowing through the capillary tube of Ubbelohde viscometer (diluted) was measured with a stopwatch. Then the original sample solution was diluted by $1 \, mol \, l^{-1}$ NaCl aqueous solution several times and the flow time of sample solutions with different dilutions was obtained. Also the flow time of $1 \, mol \, l^{-1}$ NaCl aqueous solution was measured by the same method as above. Finally, the intrinsic viscosity of samples was calculated by the dilution extrapolation method and using the Mark-Houwink equation with experimental data [43]:

$$\eta_r = \frac{t}{t_0}, \tag{2.1}$$

$$\eta_{sp} = \frac{t - t_0}{t_0} \tag{2.2}$$

and

$$[\eta] = \frac{H}{C}. \tag{2.3}$$

where $t$ and $t_0$ were the flow time of copolymer solution and $1.0 \, mol \, l^{-1}$ NaCl solution (s), $\eta_{sp}$ and $\eta_r$ were the specific viscosity and relative viscosity, $C$ was the concentration of copolymer solution ($0.0006 \, g \, ml^{-1}$), and $H$ was the mean of the $y$-intercepts of the two corresponding fitting straight lines. $[\eta]$ was the intrinsic viscosity ($ml \, g^{-1}$).

## 2.7. Rheology measurement

The correlation between the apparent viscosity and solution concentration, salinity and temperature was acquired by a Brookfield viscometer (DVII + Pro, America) and the test was carried out at the rotor speed of 20 r.p.m. The rheological behaviour of the copolymer was measured by a physical MCR 301 rotated rheometer (Austria). The shear resistance and viscoelasticity of the sample solution was tested under steady and oscillatory shear mode respectively. And all solutions of samples used for test were prepared with deionized water under room temperature conditions. The concentration of the sample solution was $6000 \, mg \, l^{-1}$ in the process of the measurements of salt resistance, temperature resistance and rheological properties.

## 2.8. Interfacial tension

The measurements of interfacial tension between crude oil and the copolymer solution and the nanocomposite solution were carried on a TX-500c spinning drop interface tensiometer (Shanghai Zhongchen, China). All sample solutions were tested for 35 min at the condition of 45°C, 6000 r.

## 2.9. Emulsifying property

Equal volumes of crude oil and sample solution were measured, respectively, into a 50 ml centrifugal tube, then an electric mixer was used to stir the mixture at a speed of 5000 r for 10 min to turn the mixture into a uniform emulsion. The prepared emulsion was poured into a measuring cylinder with a plug, and then the total volume of emulsion was recorded. Afterwards, the volume of emulsion layer was recorded at regular intervals. The emulsification property of the sample solution was evaluated by the emulsification index, it could be calculated by the following formula [3,5]:

$$EI(\%) = \frac{V_e}{V_t} \times 100\%. \tag{2.4}$$

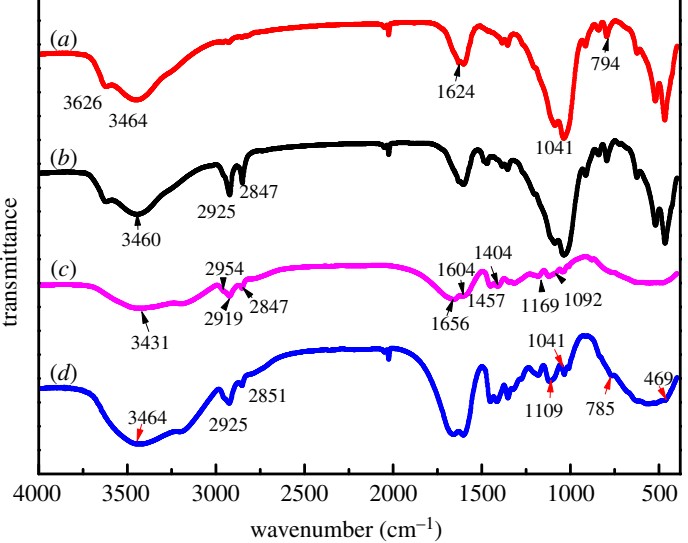

**Figure 3.** FTIR spectra of (*a*) Na-MMT, (*b*) OMMT, (*c*) ADOS (*d*) ADOS/OMMT.

In the formula above, $V_e$ is the volume of the emulsion layer, $V_t$ is the volume of total emulsion at the beginning of the test. All sample solutions were prepared using deionized water, and the whole experiment was carried out at room temperature.

# 3. Results and discussion

## 3.1. Fourier transform infrared spectra

The FTIR spectrogram of Na-MMT (figure 3*a*), OMMT (figure 3*b*), pure copolymer (figure 3*c*) and its nanocomposite (figure 3*d*) was gained and is presented in figure 3. As shown in figure 3*a*, the small peak at 3626 cm$^{-1}$ corresponded to the O—H stretching vibrations for SiO—H and AlO—H of silicate layers. The characteristic peak at 1624 cm$^{-1}$ was attributed to H—O—H bending vibrations in the water molecules, which existed in the montmorillonite interlamellar [34]. The wide band at 1041 cm$^{-1}$ and the peak value at 794 cm$^{-1}$ represented the Si—O—Si and Si—O—Al stretching vibrations separately. From the curve of figure 3*b*, the absorption bands at 2925 and 2847 cm$^{-1}$ were considered to be the symmetrical and asymmetrical telescopic vibrations of —CH$_2$ in the quaternary ammonium cationic group, respectively. In figure 3*c*, the typical absorption at 3431, 1656 and 1604 cm$^{-1}$ confirmed the presence of N—H and C=O bonds. The wide absorption band at 2919 ∼ 2954 cm$^{-1}$ and the little narrow peak at 2847 cm$^{-1}$ corresponds to the C—H stretching vibrations of —CH$_3$ and —CH$_2$—, respectively. Also, the presence of —CH3 proved the existence of the quaternary ammonium cationic group and —CH$_2$ belongs to copolymer backbone and the long chain alkyl structure. The peak at 1457 cm$^{-1}$ is the stretching vibration of N$^+$—C. The characteristic peak at 1404 cm$^{-1}$ was assigned to the benzene ring, and the absorption bands of —SO$^{3-}$ could be found at 1169 and 1092 cm$^{-1}$. The emergence of the above characteristic peaks confirmed the synthesis of copolymer (ADOS). For the curve in figure 3, the characteristic peaks that appeared in figure 3*c* reappeared in figure 3*d* too. In addition, the absorption bands of the Si—O—Al stretching vibrations and Si—O—Si bending vibration were found at 785 and 469 cm$^{-1}$, respectively in figure 3*d*, which did not appear in figure 3*c*. According to the above spectral analysis and comparison, it was certain that the nanocomposite (ADOS/OMMT) had been synthesized successfully.

## 3.2. $^1$Hydrogen nuclear magnetic resonance analysis

$^1$HNMR spectra of DOAAB and ADOS are displayed in figure 4. As shown in figure 4*a* for DOAAB, the two peaks located at 0.79 ppm and 1.24 ppm, respectively, belonged to the protons of —CH$_3$ and (—CH$_2$)$_{15}$ in the hydrophobic alkyl chain. The signal at 3.05 ppm was owing to the protons of (CH$_3$)$_2$—N—. The small peak that appeared at 3.23 ppm belonged to protons of the —CH$_2$—N— group bonded with the alkyl chain. The

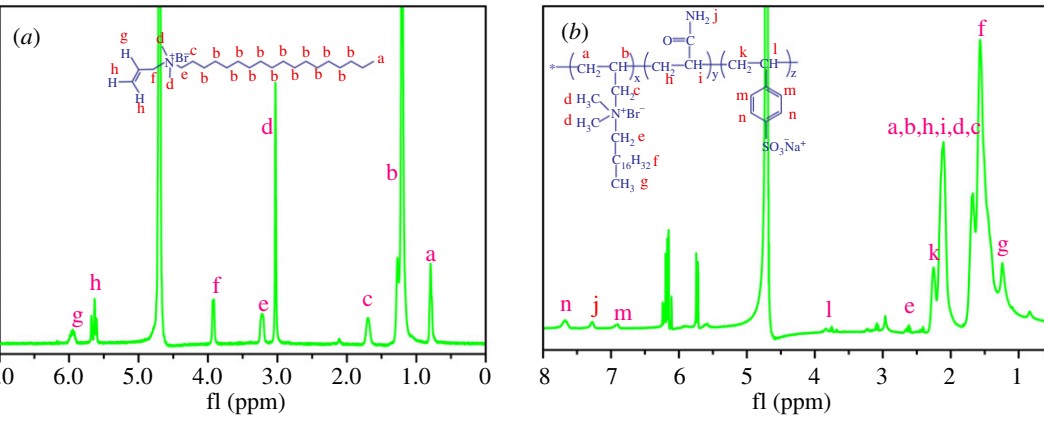

**Figure 4.** $^{1}$HNMR (in D$_2$O) spectrum of (*a*) DOAAB and (*b*) ADOS.

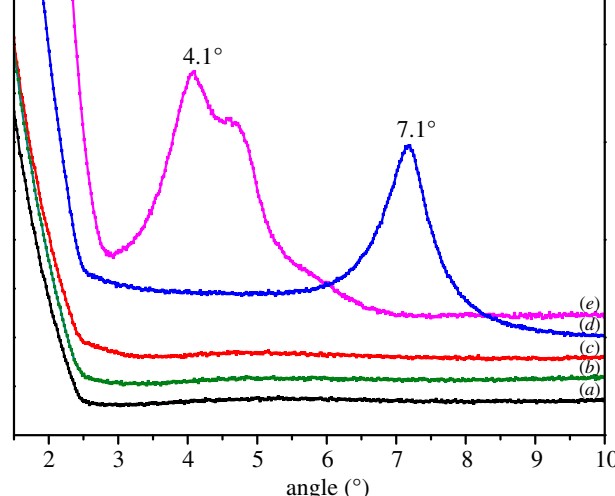

**Figure 5.** XRD pattern of (*a*) ADOS, (*b*) ADOS/1% OMMT, (*c*) ADOS/2% OMMT, (*d*) Na-MMT, and (*e*) OMMT.

protons of the —CH$_2$— group directly joined with —CH$_2$NCH$_2$CH=CH$_2$, and were associated with the small peak located at 1.69 ppm. The peak visible at 3.96 ppm was assigned to the protons of —CH$_2$—, and the —CH$_2$— was connected with the —CH=CH$_2$ group. The signals at 5.64 ppm and 5.94 ppm were linked to protons of the CH$_2$=CH— group. This information above can prove that the surfactant monomer (DOAAB) was synthesized successfully. As shown in figure 4*b* for ADOS, the peaks at 1.24 ppm and 1.57 ppm were associated with the methyl protons and methylene protons in the hydrophobic alkyl long chain, respectively. The signals of protons of methyl and methylene of —CH$_2$—N$^+$(CH$_3$)$_2$—CH$_2$ appeared at about 2.39–2.66 ppm. The corresponding peak of the protons on amide nitrogen appears at 7.28 ppm. The two peaks owing to the protons of the benzene ring were observed at 7.67 and 6.69 ppm (n, m), and the characteristic peaks of protons in the polymer skeleton appeared at 2.11–2.25 and 3.75 ppm, respectively (a,b,h,i,l,k). The above spectrum information together indicates the successful synthesis of the desired copolymer ADOS, which was consistent with the FTIR spectra information of the copolymer ADOS.

## 3.3. X-ray diffraction analysis

The modification effect of pure Na-MMT and the degree of exfoliation and intercalation of OMMT in the polymer matrix were characterized by XRD. The XRD patterns of pure Na-MMT, OMMT, pure copolymer (ADOS) and nanocomposite (ADOS/OMMT) are presented in figure 5. From the XRD curve of figure 5*d*, the diffraction peak of pure Na-MMT can be seen at 7.1°(2$\theta$). Compared with the former, the diffraction peak of OMMT (figure 5*e*) shifted forward to 4.1°(2$\theta$). Calculating by Bragg's equation indicates that the layer spacing of the Na-MMT and OMMT was 1.24 nm and 2.15 nm, respectively. The above analysis suggests that the layer spacing of the modified MMT enlarged significantly, and OMMT was prepared successfully. In figure 5*a,b* and *c*, no MMT diffraction peaks

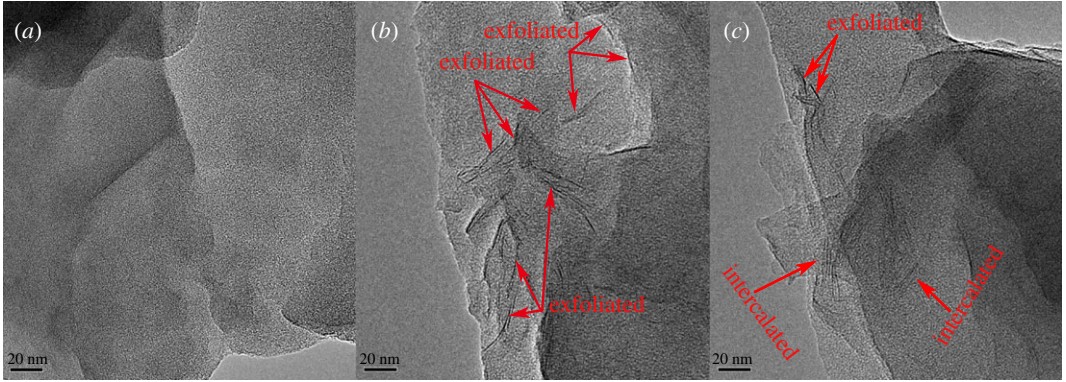

**Figure 6.** TEM images of (*a*) ADOS copolymer (at 20 nm), (*b*) ADOS/1% OMMT nanocomposite (at 20 nm), and (*c*) ADOS/2% OMMT nanocomposite (at 20 nm).

appeared in the XRD patterns of the copolymer (ADOS) and nanocomposite (ADOS/OMMT). These phenomena showed that the ADOS chains were well intercalated into the interlayers of OMMT [31]. Also, OMMT tactoids were dispersed into matrix of the ADOS by exfoliation and intercalation structures.

## 3.4. Transmission electron microscopy analysis

To observe the distribution of OMMT in the copolymer matrix intuitively, the microstructures of the samples were analysed by TEM. TEM micrographs of pure copolymer (ADOS) and nanocomposite (ADOS/1%-2% OMMT) are shown in figure 6. The TEM image (figure 6*a*) of pure copolymer (ADOS) was white and transparent owing to the penetration of electron beams. Figure 4*b* and *c* show TEM images of nanocomposite, in which black lines appear in the brighter region. Black lines correspond to MMT platelets while the brighter region correspond to the copolymer matrix, which was because, compared to the copolymer matrix composed of C, H and N, the MMT were made up of the heavier elements O, Si and Al [44]. In the TEM micrograph (figure 6*b*) of the nanocomposite with 1.0 wt% OMMT, exfoliated nanolayers of MMT dispersed randomly in the copolymer matrix. However, exfoliated MMT platelets and intercalated structure existed, when nanocomposite was 2.0 wt% OMMT (figure 6*c*). This phenomenon showed that overloading of OMMT was detrimental for exfoliation of OMMT layers owing to adhesion of the nanostructures.

## 3.5. Thermo-gravimetric analyser analysis

The effects of the introduction of OMMT on the thermal stability properties of copolymers was studied using TGA. The TGA curves of ADOS and ADOS/OMMT are exhibited in figure 7.

Obviously, both the pure copolymer and the nanocomposites showed three-step degradation [45]. At the first thermal degradation stage, the degradation initial temperature of pure copolymer and the nanocomposite samples was much the same. At the second degradation stage, the decomposing temperature increased from 243.6°C for pure copolymer to 251.3°C and 257.8°C for the ADOS/1% OMMT and ADOS/2% OMMT nanocomposites, respectively. At the third degradation stage, that is, the stage of fastest mass loss, the decomposing temperature ($T_{max}$) of pure copolymer, the ADOS/1% OMMT and ADOS/2% OMMT nanocomposites were 366.5°C, 374.3°C and 379.1°C, respectively. Quite obviously, the thermal degradation temperature of ADOS/OMMT nanocomposite was higher than that of pure ADOS. Moreover, it can be clearly observed in the figure that the mass loss of pure copolymer was more than that of nanocomposites in the same thermal degradation stage. Therefore, ADOS/OMMT nanocomposite had better thermal stability compared to pure ADOS, especially nanocomposite containing 2 wt% OMMT. These results were related to the gas and heat barrier action of OMMT. The OMMT layers dispersed in the polymer matrix not only hindered the contact between oxygen and the polymer, but also slowed down the heat transfer in the matrix at high temperature, thus reducing the decomposition rate of the polymer, and delaying the volatilization of the decomposition products [46–48].

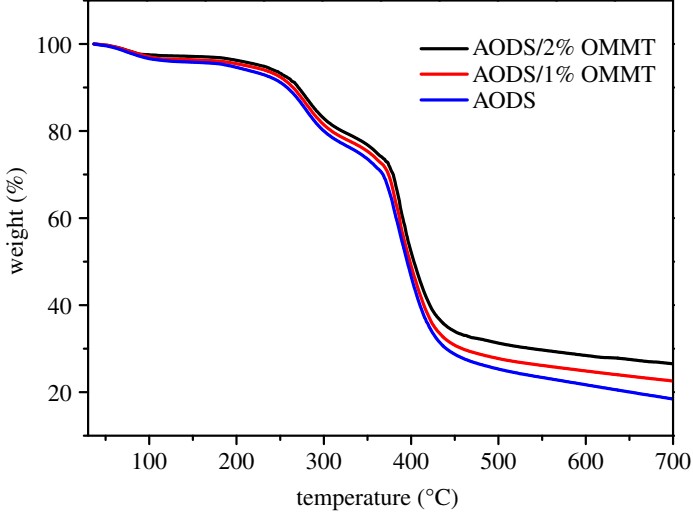

**Figure 7.** TGA curves of ADOS, ADOS/1% OMMT and ADOS/2% OMMT nanocomposites.

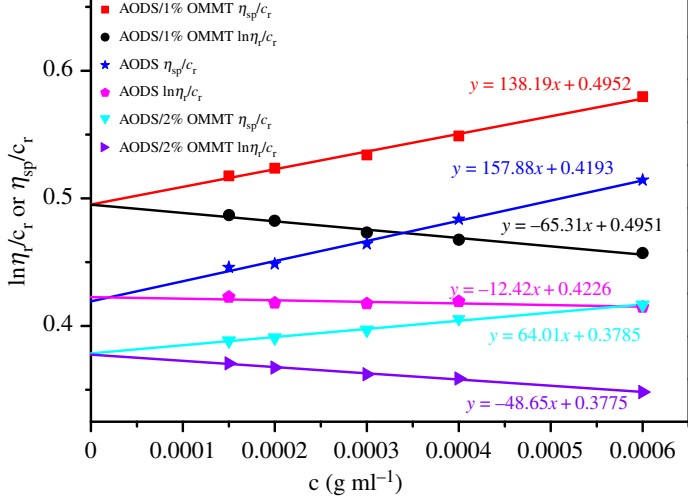

**Figure 8.** The function relationship between $\eta_{sp}/c_r$ or $\ln\eta_r/c_r$ and $c$ (samples).

## 3.6. Intrinsic viscosity

The intrinsic viscosity $[\eta]$ of the copolymer sample reflects the contribution of an individual copolymer molecule to the solution viscosity and it doesn't change with the concentration of sample solution [49,50]. As figure 8 shows, taking $\ln\eta_r/c_r$ and $\eta_{sp}/c_r$ as the ordinate and the practical concentration of sample solution as the abscissa, the function diagram between $\ln\eta_r/c_r$ and $\eta_{sp}/c_r$ and the corresponding concentration was acquired by linear fitting. The $[\eta]$ of ADOS, ADOS/1% OMMT and ADOS/2% OMMT nanocomposites were respectively 701.583 ml g$^{-1}$, 825.25 ml g$^{-1}$, 630.00 ml g$^{-1}$ using dilution extrapolation.

## 3.7. Apparent viscosity

In order to study the difference between the influence trend of the concentration of the nanocomposites solutions and the pure copolymer solution on the apparent viscosity, the apparent viscosity of the pure polymer solution and the nanocomposites solutions with different concentrations was tested at room temperature, as shown in figure 9. It can be observed from the figure that all three curves showed an apparent turning point when the concentration was about 3000 mg l$^{-1}$, which was the critical micelle concentration (CMC) of the samples. Obviously, the apparent viscosity increased slowly as the polymer concentration increased before the turning point. This was because at low concentrations, the

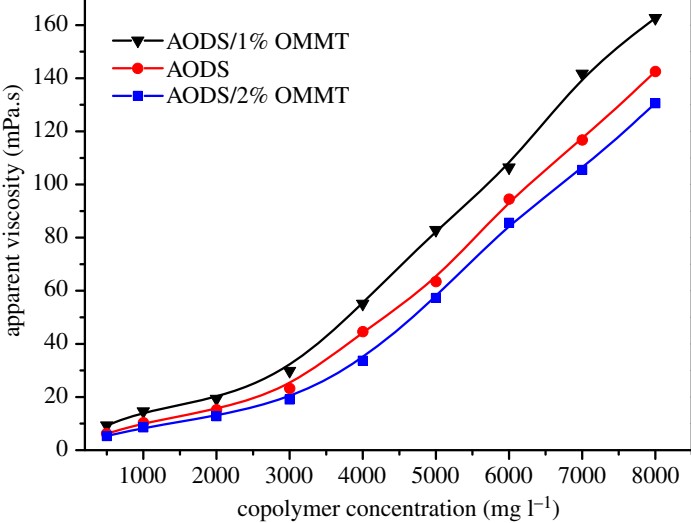

**Figure 9.** Relationship between apparent viscosity and samples concentration ($T = 25°C$).

polymer macromolecules were mainly in the form of intramolecular association in the solution, which led to the curl of the molecular chain and the reduction of the hydrodynamic volume. After the turning point, intermolecular association played a dominant role, and supramolecular aggregates with network structure formed, leading to the sharp increase of apparent viscosity [51]. However, at the same concentration, the apparent viscosity of the ADOS/1% OMMT nanocomposite solution was higher than the OMMT-free one, and the viscosity increased more remarkably after the critical micelle concentration, while the ADOS/2% OMMT nanocomposite solution did the opposite. The reasons for these results could be interpreted as follows. When the additional amount of OMMT was 1 wt%, the OMMT was highly exfoliated and the exfoliated OMMT layer acted as a bridge, which enhanced the intermolecular association, so higher apparent viscosities resulted, forming a more complex network structure [44,52]. However, when the additional amount of OMMT was 2 wt%, the OMMT was mostly in the form of an intercalation structure which may hinder the growth of the polymer chain, weaken the intermolecular association and the stability of the network structure, leading to the decrease of viscosity.

## 3.8. Temperature-tolerance performance

The influence of temperature from 25°C to 95°C on viscosity of ADOS solution and ADOS/OMMT solution for the concentrations of 6000 mg l$^{-1}$ is shown in figure 10.

It can be seen from the figure 10, the apparent viscosity of both the ADOS solution and ADOS/OMMT solution showed a decreasing trend with increasing temperature. It meant that rising temperatures had an adverse impact on intermolecular hydrophobic interactions, leading to the weakening of physical network cross-linking [14,16,53]. Interestingly, however, nanocomposite solutions had a more gentle viscosity-temperature curve than pure polymer solutions and when the temperature rose to 95°C, the viscosity retention rates of ADOS/1% OMMT and ADOS/2% OMMT were 38.36% and 28.03%, respectively, while the viscosity retention rates of ADOS was only 24.32%. Therefore, compared with pure polymers (ADOS), ADOS/OMMT had better temperature resistance. These results indicated that the OMMT layers dispersed in the polymer matrix could effectively reduce the destruction of temperature to the intermolecular hydrophobic association network.

## 3.9. Salt-resistant properties

Salt ions are known to affect the apparent viscosity of polymer solutions. Therefore, the apparent viscosity in the presence of different concentrations of NaCl and Ca$_2$Cl for both ADOS and ADOS/OMMT solutions was tested and recorded. The apparent viscosity of the sample solution as a function of different salt ions (NaCl and Ca$_2$Cl) concentration is summarized in figure 11. It was obvious that all viscosity curves in figure 11 have similar trends. That is, with the increase of NaCl and CaCl$_2$ concentrations, the apparent viscosities of the samples solutions decreased first, then increased, and

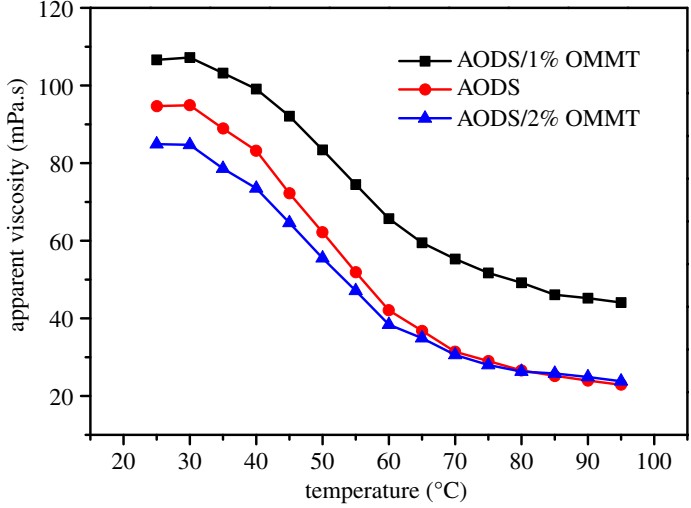

**Figure 10.** Effect of temperature on apparent viscosity of samples solutions.

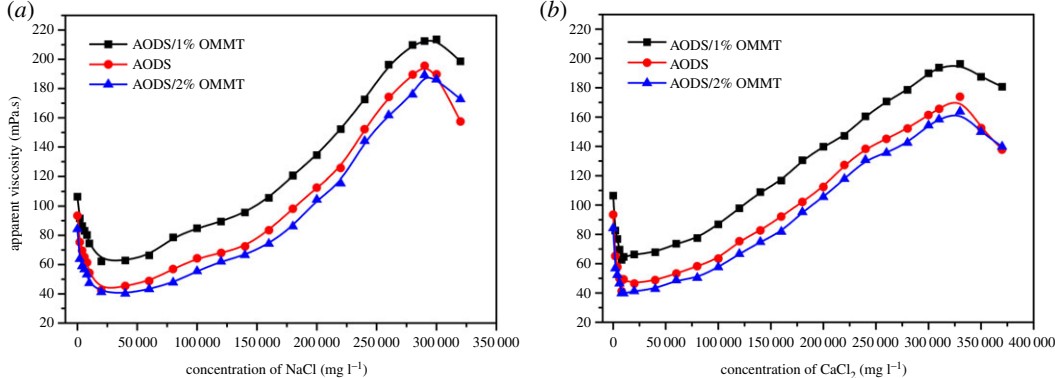

**Figure 11.** Effect of (*a*) NaCl and (*b*) CaCl$_2$ concentration on apparent viscosity of sample solutions ($T = 25$℃).

then decreased. And similar phenomenon could be seen in previous literature [54]. These changing trends in apparent viscosity could be explained as follows. At low salt concentration a shielding effect played a leading role. The static charges (—COO$^-$ and —SO$_3$H$^-$) on the polymer backbone were shielded by salt ions (Na$^+$ or Ca$^{2+}$), leading to the reduction of electrostatic repulsion and the contraction of polymer chains. This resulted in the decline of the sample's apparent viscosities [55,56]. At the intermediate salt concentration, solvents polarity enhanced owing to increasing salt concentration. This environment was conducive to the improvement of intermolecular hydrophobic association, thus forming more cross-link networks and larger intermolecular aggregates, so, viscosity increment occurred [57,58]. At high salt concentration, cross-link networks dissociated owing to the increase of electrostatic repulsion between molecular chains, so the sample's apparent viscosity decreased again [51]. However, at low salt concentration and high salt concentration, it can be observed that the viscosity of ADOS/OMMT solution decreases more slowly than that of pure polymer solution with the increase of salt concentration. Moreover, the lowest viscosity retention of ADOS/1% OMMT and ADOS/2% OMMT solution are higher than that of pure polymer solution in different salt solutions. These results may be explained as follows. The MMT platelets are a protective barrier that somewhat reduces the shielding effect of Na$^+$ and Ca$^{2+}$ on the charge on the polymer chain. In addition, MMT platelets could promote and participate in formation of stronger intermolecular association networks [27]. Also, as can be seen from figure 11, at low salt concentration, the minimum viscosity retention rate of the samples in both NaCl and CaCl$_2$ solutions exceeds 40%. Moreover, when the concentration of NaCl and CaCl$_2$ solutions was about 290 000 mg l$^{-1}$ and 330 000 mg l$^{-1}$, respectively, the apparent viscosity of the samples reached the maximum value. Above all, both ADOS and ADOS/OMMT nanocomposites had excellent salt resistance, and the introduction of OMMT has a synergistic effect on the salt resistance of pure polymer.

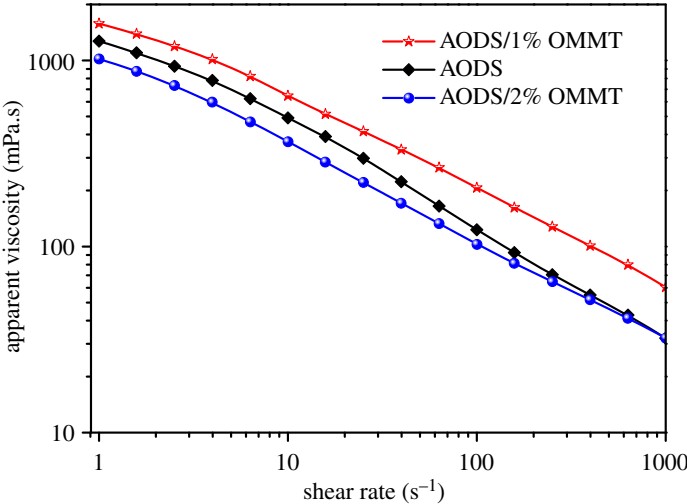

**Figure 12.** Apparent viscosity of ADOS, ADOS/1% MMT and ADOS/2% MMT as a function of shear rate ($T = 25°C$).

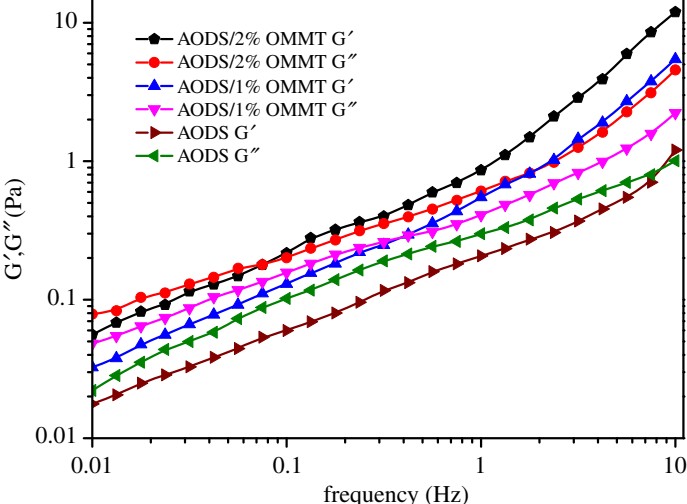

**Figure 13.** The dynamic moduli of ADOS, ADOS/1% MMT and ADOS/2% MMT as a function of frequency ($T = 25°C$).

## 3.10. Shear-resistant properties

Variation curves of ADOS and ADOS/OMMT solutions apparent viscosity versus shear rate are demonstrated in figure 12. As shown in figure 12, as the shear rate increased from $1\,s^{-1}$ to $1000\,s^{-1}$, the apparent viscosity of the sample solution decreased gradually, showing the behaviour of shear thinning. However, under the same shearing rate, the shear degradation of ADOS/1% OMMT and ADOS/2% OMMT was lower as compared with that for ADOS. This was because the introduction of nano-OMMT enhanced the intermolecular hydrophobic association, which made the three-dimensional network structure more stable and less vulnerable to shear damage. Consequently, ADOS/OMMT nanocomposites had more excellent shear resistance performance than ADOS.

## 3.11. Viscoelastic behaviour

The viscoelastic property of the polymer solution is of great significance to the displacement of oil trapped in pore dead angles. The viscoelasticity test results of ADOS and ADOS/OMMT solutions are reported in figure 13. With the oscillation frequency rising up, the storage moduli (G′) and the loss moduli (G″) of all samples presented a trend of increasing gradually. And it was obvious that the G′ and G″ of ADOS/OMMT are always greater than those of ADOS showing more superior viscoelasticity. Besides, the G′ and G″ of all samples crossed at their respective critical frequency

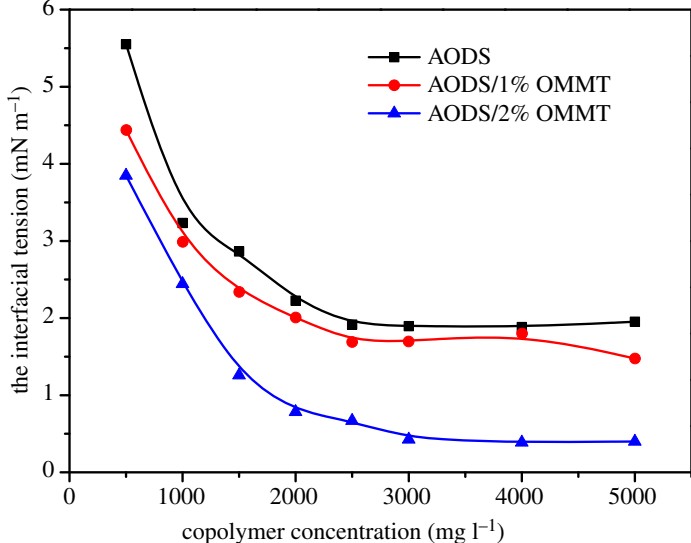

**Figure 14.** Effect of ADOS, ADOS/1% MMT and ADOS/2% MMT concentrations on crude oil–water IFTs.

values, and above the critical frequency (fc), G′ are larger than G″, which indicated elasticity played a dominating role. Meanwhile, the reciprocal of the frequency ($f_c$) corresponding to the crossing point ($G_c$) was the characteristic relaxation time ($t_c$) [59]. Therefore, the $t_c$ of ADOS, ADOS/1% OMMT and ADOS/2% OMMT were 0.12, 2.7 and 13.7 s, respectively. Moreover, the longer $t_c$ was, the better the elastic efficiency. So, compared with ADOS, ADOS/OMMT had better elastic properties. These results could be interpreted as, the original intermolecular hydrophobic association was strengthened by electrostatic interaction and hydrogen bonding between molecular chains of matrix and exfoliated MMT layers, forming a more complex and stable physical network structure [60].

## 3.12. Interfacial tension measurement

Interfacial tension (IFT) of both ADOS and ADOS/OMMT nanocomposites at the oil–water interface have been tested. The relation curves between IFT of ADOS and ADOS/OMMT nanocomposite solution and their solution concentration at 45°C are demonstrated in figure 14. It can be seen that the IFT of solution of both ADOS and ADOS/OMMT nanocomposites decreased sharply first and then changed slightly with the rising of solution concentration of the sample. Furthermore, the curves of ADOS and ADOS/OMMT nanocomposites appeared as inflection points on their respective curves. The sample concentrations of ADOS, ADOS/1% OMMT and ADOS/2% OMMT nanocomposites at the inflection points were 2500, 2500 and 3000 mg l$^{-1}$, respectively. Furthermore, the ADOS and ADOS/OMMT nanocomposite had a relatively good ability to decrease the interfacial tension between crude oil and water. Interestingly, ADOS/OMMT nanocomposites could contribute to more decrease than ADOS in the interfacial tension at the crude oil–water interface. These phenomena attributed to the fact that the hydrophilic and hydrophobic segments existing in copolymers had good orientation at the crude oil–water interface, which led to reduction of interfacial tension [18,61]. Additionally, the introduced nanoparticles of OMMT may modify the interfacial properties of crude oil–water systems, also, there might be a good synergy between nanoparticle and copolymer. Hence, the ADOS/OMMT nanocomposites solution showed better interfacial activity, as compared with pure polymer ADOS [3,62].

## 3.13. Emulsifying ability

The emulsifying capacity of the ADOS and ADOS/OMMT nanocomposites at room temperature was investigated by recording emulsification index against time. The results of emulsifying crude oil with these samples in aqueous solution are displayed in figure 15. Clearly, whether the concentration of these samples was 4000 mg l$^{-1}$ or 5000 mg l$^{-1}$, the emulsification index of ADOS/1% OMMT nanocomposites was greater than pure polymer and ADOS/2% OMMT nanocomposites throughout the test, that is,

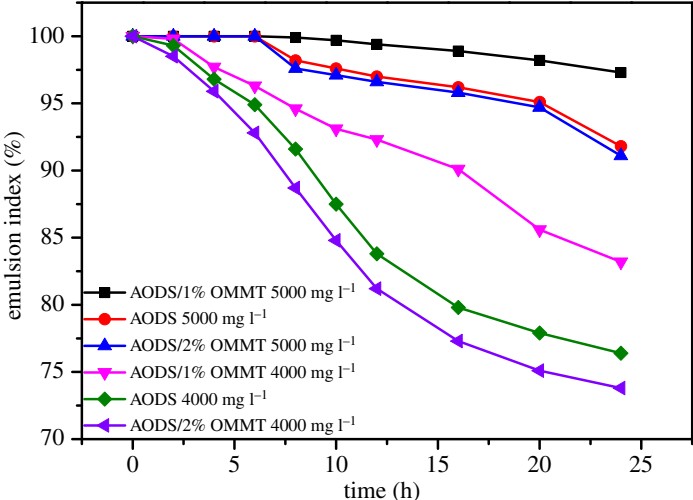

**Figure 15.** The ability of samples to stabilize emulsions at different concentrations (the volume ratio of crude oil and water is 1 : 1).

the emulsifying capacity of ADOS/1% OMMT nanocomposites was more superior than the other two samples. Moreover, it could be noticed that pure polymers had a better ability to stabilize emulsions in aqueous solutions than ADOS/2% OMMT nanocomposites. This was because the amphiphilic structure of the copolymer gave the copolymer the ability to emulsify crude oil. The hydrophobic groups of the copolymer could form network structure by intermolecular association, which could trap oil droplets into network structure thereby forming stable O/W emulsion with the interface membrane [63]. Additionally, the incorporation of appropriate amounts of MMT layers contributed to the stability of emulsion, which may be that the exfoliated OMMT layers could enhance the stability of the network structure on the surface of oil droplets [64]. Nevertheless, when the amount of MMT exceeded the right amount, the molecular weight of the polymer decreased, resulting a viscosity reduction, and then emulsion stability became poor.

# 4. Conclusion

A amphoteric hydrophobically associative polyacrylamide/organic MMT nanocomposite (ADOS/OMMT) was prepared with *in situ* polymerization. TEM and XRD analysis proved that OMMT was successfully dispersed in an ADOS copolymer matrix in the form of exfoliation and intercalated structure. The TGA results showed that the thermal stability of the ADOS/OMMT was better than pure ADOS. Meanwhile, the solution properties investigation indicated that ADOS/OMMT had higher apparent viscosity than pure ADOS. Besides, at the same condition, the temperature resistance, shear resistance and viscoelasticity of the ADOS/OMMT solution were enhanced relative to ADOS. Also, compared with pure ADOS, the copolymer ADOS with introduction of an appropriate amount of OMMT could demonstrate superior interfacial activity and crude oil emulsifiability. Therefore, ADOS/OMMT nanocomposites possess good application prospects in tertiary recovery and oil drilling.

Data accessibility. All the necessary data are included in the main manuscript and figures, and raw data for all the figures in the paper have been uploaded as the electronic supplementary material.

Authors' contributions. C.B. designed this study, carried out the laboratory experiments and wrote the manuscript. X.H., Y.K. and L.X. contributed to the decision of the experimental plan and revised the manuscript. Y.Z. and S.L. participated in the data analysis. Y.L. assisted in the testing experimental work. All authors gave final approval for publication.

Competing interests. We declare we have no competing interests.

Funding. This work was financially supported by the National Natural Science Foundation of China (grant nos. 51974339 and 51674270), National Major Project (grant no. 2017ZX05009-003), Major project of the National Natural Science Foundation of China (grant no. 51490650) and the Foundation for Innovative Research Groups of the National Natural Science Foundation of China (grant nos. 51821092 and 51521063).

Acknowledgements. We would like to thank Ma T. and G SX. for their assistance with rheological properties tests.

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
