## [Reviewer comments · Royal Society Open Science]

Review History

RSOS-200199.R0 (Original submission)

Review form: Reviewer 1

Is the manuscript scientifically sound in its present form?

Yes

Are the interpretations and conclusions justified by the results?

Yes

Is the language acceptable?

Yes

Do you have any ethical concerns with this paper?

No

Have you any concerns about statistical analyses in this paper?

No

Recommendation?

Accept with minor revision (please list in comments)

Comments to the Author(s)

This is an interesting work about the preparation and characterization of amphoteric hydrophobically associative polyacrylamide/organic MMT nanocomposite with in situ polymerization. The science is solid and the characterization is complete. However, I have some minor concerns before I can recommend it for publication.

(1) The main new insight of this work focuses on preparation and characterization of materials. Limited new insight is provided with respect to application of materials. I would like to suggest include some real application of materials.

(2) In Figure 6, the scale bar is a little small. I would suggest the authors make it bigger.

Based on the above concerns, I suggest a minor revision.

Review form: Reviewer 2

Is the manuscript scientifically sound in its present form?

Yes

Are the interpretations and conclusions justified by the results?

Yes

Is the language acceptable?

Yes

Do you have any ethical concerns with this paper?

No

Have you any concerns about statistical analyses in this paper?

No

Recommendation?

Accept as is

Comments to the Author(s)

Author prepared an amphoteric hydrophobically associative polyacrylamide/organic MMT nanocomposite (ADOS/OMMT) by in situ polymerization. TEM and XRD analysis proved that OMMT was successfully dispersed in ADOS copolymer matrix in the form of exfoliation and intercalated structure. The TGA results showed that the thermal stability of the ADOS/OMMT was better than pure ADOS. The prepared ADOS/OMMT showed the higher apparent viscosity than pure ADOS. Additionally, the temperature resistance, shear resistance and viscoelasticity of ADOS/OMMT solution were enhanced relative to ADOS. Finally, the prepared ADOS/OMMT composite showed better performance like interfacial activity and crude oil emulsifiability than that of the pure ADOS. ADOS/OMMT nanocomposites possess good application prospects in tertiary recovery and oil drilling. The synthesis and characterization of the materials were done perfectly. The prepared amphoteric hydrophobically associative polyacrylamide/organic MMT nanocomposite (ADOS/OMMT) is also novel. So, the paper can be published as it is.

Decision letter (RSOS-200199.R0)

13-Mar-2020

Dear Professor Ke:

Title: Preparation and properties of amphiphilic hydrophobically associative polymer/
montmorillonite nanocomposites
Manuscript ID: RSOS-200199

Thank you for submitting the above manuscript to Royal Society Open Science. On behalf of the Editors and the Royal Society of Chemistry, I am pleased to inform you that your manuscript will be accepted for publication in Royal Society Open Science subject to minor revision in accordance with the referee suggestions. Please find the reviewers' comments at the end of this email.

The reviewers and handling editors have recommended publication, but also suggest some minor revisions to your manuscript. Therefore, I invite you to respond to the comments and revise your manuscript.

Because the schedule for publication is very tight, it is a condition of publication that you submit the revised version of your manuscript before 22-Mar-2020. Please note that the revision deadline will expire at 00.00am on this date. If you do not think you will be able to meet this date please let me know immediately.

Supplementary files will be published alongside the paper on the journal website and posted on the online figshare repository (<https://figshare.com>). The heading and legend provided for each

supplementary file during the submission process will be used to create the figshare page, so please ensure these are accurate and informative so that your files can be found in searches. Files on figshare will be made available approximately one week before the accompanying article so that the supplementary material can be attributed a unique DOI.

Best wishes,
Dr Laura Smith
Publishing Editor, Journals

RSC Associate Editor:
Comments to the Author:
(There are no comments.)

RSC Subject Editor:
Comments to the Author:
(There are no comments.)

Reviewer comments to Author:
Reviewer: 1

Comments to the Author(s)

This is an interesting work about the preparation and characterization of amphoteric hydrophobically associative polyacrylamide/organic MMT nanocomposite with in situ polymerization. The science is solid and the characterization is complete. However, I have some minor concerns before I can recommend it for publication.

(1) The main new insight of this work focuses on preparation and characterization of materials. Limited new insight is provided with respect to application of materials. I would like to suggest include some real application of materials.

(2) In Figure 6, the scale bar is a little small. I would suggest the authors make it bigger.

Based on the above concerns, I suggest a minor revision.

Reviewer: 2

Comments to the Author(s)

Author prepared an amphoteric hydrophobically associative polyacrylamide/organic MMT nanocomposite (ADOS/OMMT) by in situ polymerization. TEM and XRD analysis proved that OMMT was successfully dispersed in ADOS copolymer matrix in the form of exfoliation and intercalated structure. The TGA results showed that the thermal stability of the ADOS/OMMT was better than pure ADOS. The prepared ADOS/OMMT showed the higher apparent viscosity than pure ADOS. Additionally, the temperature resistance, shear resistance and viscoelasticity of ADOS/OMMT solution were enhanced relative to ADOS. Finally, the prepared ADOS/OMMT composite showed better performance like interfacial activity and crude oil emulsifiability than that of the pure ADOS. ADOS/OMMT nanocomposites possess good application prospects in tertiary recovery and oil drilling. The synthesis and characterization of the materials were done perfectly. The prepared amphoteric hydrophobically associative polyacrylamide/organic MMT nanocomposite (ADOS/OMMT) is also novel. So, the paper can be published as it is.

Author's Response to Decision Letter for (RSOS-200199.R0)

See Appendix A.

Decision letter (RSOS-200199.R1)

15-Apr-2020

Dear Professor Ke:

Title: Preparation and properties of amphiphilic hydrophobically associative polymer/
montmorillonite nanocomposites
Manuscript ID: RSOS-200199.R1

It is a pleasure to accept your manuscript in its current form for publication in Royal Society Open Science. The chemistry content of Royal Society Open Science is published in collaboration with the Royal Society of Chemistry.

RSC Associate Editor
Comments to the Author:
The manuscript can be accepted as is.

Reviewer(s)' Comments to Author:

Appendix A

Dear Editor Dr Laura Smith,

Thanks very much for your letter and the reviewers' comments on this manuscript entitled "Preparation and properties of amphiphilic hydrophobically associative polymer/ montmorillonite nanocomposites" (Manuscript ID: RSOS-200199). These comments are all valuable and very helpful for revising and improving our paper, as well as the important guiding to our researches in the future. We have carefully studied the comments and made revisions which we hope meet with approval. Once again, we sincerely appreciate Editors/Reviewers' warm work earnestly. The main revisions and responds to the reviewer's comments are as follows:

To reviewer #1:

Comments to the Author(s)

This is an interesting work about the preparation and characterization of amphoteric hydrophobically associative polyacrylamide/organic MMT nanocomposite with in situ polymerization. The science is solid and the characterization is complete. However, I have some minor concerns before I can recommend it for publication.

(1) The main new insight of this work focuses on preparation and characterization of materials. Limited new insight is provided with respect to application of materials. I would like to suggest include some real application of materials.

Response 1: We greatly thank you for raising the good suggestion and giving guidance. This paper is mainly about the preparation and characterization of amphiphilic hydrophobically associative polymer/ montmorillonite nanocomposites. Regarding the real application of materials, our relevant work is in progress and will be reported later.

(2) In Figure 6, the scale bar is a little small. I would suggest the authors make it bigger.

Response 2: Thanks for your nice advice. We have cropped out the extra white space in the previous picture (Figure 6) and increased the aspect ratio of the picture (Figure

6) to make it bigger. The revised picture is as follows (Also, we have put the revised picture in the revised manuscript (in page 34). And the captions of revised picture are highlighted in blue in the revised manuscript, please check it.):

Figure 6. TEM images of (a) ADOS copolymer (at 20 nm), (b) ADOS/1% OMMT nanocomposite (at 20 nm), (c) ADOS/2% OMMT nanocomposite (at 20 nm).

To reviewer #2:

Comments to the Author(s)

Author prepared an amphoteric hydrophobically associative polyacrylamide/organic MMT nanocomposite (ADOS/OMMT) by in situ polymerization. TEM and XRD analysis proved that OMMT was successfully dispersed in ADOS copolymer matrix in the form of exfoliation and intercalated structure. The TGA results showed that the thermal stability of the ADOS/OMMT was better than pure ADOS. The prepared ADOS/OMMT showed the higher apparent viscosity than pure ADOS. Additionally, the temperature resistance, shear resistance and viscoelasticity of ADOS/OMMT solution were enhanced relative to ADOS. Finally, the prepared ADOS/OMMT composite showed better performance like interfacial activity and crude oil emulsifiability than that of the pure ADOS. ADOS/OMMT nanocomposites possess good application prospects in tertiary recovery and oil drilling. The synthesis and characterization of the materials were done perfectly. The prepared amphoteric hydrophobically associative polyacrylamide/organic MMT nanocomposite (ADOS/OMMT) is also novel. So, the paper can be published as it is.

Response: Thank you for your careful review and full recognition of our manuscript.

We would be very happy to provide further information if required, and we will greatly appreciate your great contribution to our manuscript.

Yours Sincerely,

Yangchuan Ke

CNPC Nano Chemistry Key Laboratory, College of Science, China University of Petroleum, Beijing, China.

Email address: kyc031@sohu.com